# The Impact of Growing Area on the Expression of Fruit Traits Related to Sensory Perception in Two Tomato Cultivars

**DOI:** 10.3390/ijms25169015

**Published:** 2024-08-20

**Authors:** Daniela D’Esposito, Antimo Di Donato, Sharon Puleo, Matteo Nava, Gianfranco Diretto, Rossella Di Monaco, Luigi Frusciante, Maria Raffaella Ercolano

**Affiliations:** 1Department of Agricultural Sciences, University of Naples Federico II, 80055 Portici, Italy; daniela.desposito@unina.it (D.D.); antimo.didonato@unina.it (A.D.D.); sharon.puleo@unina.it (S.P.); rossella.dimonaco@unina.it (R.D.M.); fruscian@unina.it (L.F.); 2Italian National Agency for New Technologies, Energy and Sustainable Development (ENEA), Casaccia, 00123 Rome, Italy; matteo.nava@enea.it (M.N.);

**Keywords:** environment, fruit quality, metabolite, sensory attributes, *Solanum lycopersicum*, transcriptome

## Abstract

Environmental conditions greatly influence the quality of tomato fruit by affecting the expression of genes, the abundance of metabolites, and the perception of sensorial attributes. In this study, a fruit transcriptome investigation, a sensory test, and a metabolomic analysis were performed to evaluate the impact of the environment on two popular tomato cultivars grown in two Italian regions. The transcriptional profile of each cultivar, cultivated in two different areas, highlighted differential expression in genes involved in pathways related to cell wall components such as pectin, lignin, and hemicellulose and sugars as well as in amino acids, phenylpropanoids, and pigment synthesis. The cultivation area mainly affects sensory attributes related to texture and flavor and the metabolic pattern of cell wall precursors, sugars, glutamate, aspartate, and carotenoids. In the two genotypes cultivated in the same environment, some attributes and fruit-related quality processes are similarly affected, while others are differently influenced based on the specific genetic makeup of the tomato. A combination of transcriptomic, sensory, and metabolomic data obtained from the two tomato genotypes revealed that the environment has a profound effect on specific sensory traits, providing information on factors that shape the specific characteristics and genetic targets for improving tomato fruit characteristics.

## 1. Introduction

The quality of tomatoes is a complex trait regulated by many factors. In particular, sensory perception is driven by the combination of taste, texture, and smell attributes. Intense taste is the result of an increase in gluconeogenesis, the hydrolysis of polysaccharides, a decrease in acidity, and the accumulation of sugars and organic acids [1], while the aroma is produced by a complex mixture of volatile compounds and the degradation of bitter principles, flavonoids, tannins, and related compounds [1,2]. The color of the fruits is mainly determined by carotenoids and flavonoids [3,4], while the texture is primarily due to the cell wall structure but, also, the cuticle and the morphology of the fruit can have effects on it [5]. In recent years, the characteristics of tomato fruits related to sensory perception have been extensively studied at genetic and biochemical levels to obtain new cultivars with improved taste [6,7].

Climate fluctuations, due to land transformation, gas emissions into the atmosphere, and other human-made activities, have rapidly increased in the last 20 years [8]. Therefore, nowadays it is important to select tomato cultivars resilient to cultivation perturbations with good production and quality traits. Consumers have expectations about the sensory profile of tomatoes, and sensory traits are more pleasant if they confirm the expected level of quality. In this context, the overall flavor and firmness are the most important traits for improving tomato fruit quality [9].

Genomic studies are contributing in many ways to the molecular dissection of quality traits. Recently, a tomato comparative transcriptomic analysis allowed us to identify the key genes responsible for fruit sensorial quality in three tomato cultivars grown in southern Italy [10]. Specifically, the study highlighted a complex network in which plant cell components are the main hubs, buffering tomato sensory attributes and metabolic traits in different environmental conditions.

The main aim of this study was to investigate the transcriptional profile of two tomato cultivars (Dardo and Pixel) cultivated in two different areas, Battipaglia (SA) and Fondi (LT), to highlight changes in biological processes involved in fruit quality. To this end, genes involved in the metabolism of cell wall components, carbohydrates, amino acids, and other compounds involved in sensory perception were analyzed. Furthermore, the differential gene expression of the two cultivars was compared to the metabolic and sensory profile of each genotype to identify the traits mostly affected by the cultivation environment. Finally, insights on genetic targets for improving tomato quality traits were obtained.

## 2. Results

### 2.1. Pixel and Dardo Gene Expression Profile

This study allowed us to identify the gene expression profile induced in tomatoes by specific environmental conditions. In particular, the transcriptome of two tomato cultivars (Pixel and Dardo) grown in two Italian areas, Battipaglia (Ba) and Fondi (Fo), was compared. The differentially expressed genes (DEGs) for each genotype (Pixel and Dardo) were identified by comparing the gene expression levels in the two different environments using as a reference the expression profiles detected in Fondi (Figure 1A). In Dardo, 4082 DEGs were identified, of which 1925 were up-regulated and 2157 were down-regulated (Dardo Ba vs. Fo, Appendix A). On the contrary, in Pixel, 3340 genes resulted as being differentially expressed between the two areas, of which 1402 were up-regulated and 1938 were down-regulated (Pixel Ba vs. Fo, Appendix A).

The up- and down-regulated genes for both cultivars were intersected to identify specific or shared genes (Figure 1B). The cultivars shared 671 up-regulated genes and 1062 down-regulated genes (Figure 1B). From the plot, it was also possible to detect genes that had an opposite expression pattern in the two genotypes, namely 42 genes were up-regulated in Pixel and down-regulated in Dardo while 9 genes were up-regulated in Dardo and down-regulated in Pixel. Furthermore, Dardo differentially expressed 2298 specific genes (1245 up- and 1053 down-regulated), while Pixel 1557 (690 up- and 867 down-regulated). Among the genes with opposite expression in the two environments were several genes involved in fruit quality (Table 1), mainly related to carbohydrate and cell wall metabolism.

### 2.2. DEGs Involved in Primary Metabolism

To understand the functional role of the DEGs, they were mapped to different cellular compartments or processes using the MapMan tool (Figure 2). Within the primary metabolism, the environment affected both genotypes in carbohydrate and amino acid metabolism. Notably, Dardo showed 70 differentially expressed genes involved in cell wall metabolism, from them, 35 resulted in up-regulated genes (green), and 35 down-regulated (red) in Battipaglia (Figure 2A). Down-regulated genes were mainly involved in cell wall modification activities such as cellulose (cellulose synthases and cobra proteins) and xyloglucan modifications. Several up-regulated genes were detected in the pectin metabolism (pectinacetylesterases, pectinesterases) and the synthesis of hemicellulose precursors. In Pixel, out of a total of 52 DEGs identified in cell wall bins, 39 were activated in Battipaglia (Figure 2B). The hemicellulose compartment had the highest concentration of up-regulated DEGs but, also, in the pectin sector, the gene activation was predominant. Cellulose biosynthesis was activated thanks to five genes coding for cellulose synthases or cellulose-synthases-like proteins.

By focusing on the cell wall precursor synthesis, the nucleotide sugar metabolism was affected by the environment in both genotypes with a high number of up-regulated genes (Figure 3). Pixel activated the conversion between glucuronic to galacturonic and vice versa, while Dardo repressed this conversion. In addition, Pixel up-regulated the gene *Solyc12g010540*, coding for UDP-glucuronate 4-epimerase, that in Dardo was down-regulated (see also Table 1). Instead, Dardo Battipaglia showed the presence of down-regulated genes for the conversion of UDG-glucose in UDP-galactose (and vice versa) as well as the up-regulation of a gene coding for the myo-inositol oxygenase, (*Solyc06g062430*) involved in the conversion of myo-inositol in glucuronic acid. In both genotypes, the activation of sugar nucleotide transport was observed. Dardo down-regulated two UDP-galactose transporters *(Solyc01g091840*, *Solyc01g010650*), one of which was up-regulated in Pixel too (*Solyc01g091840*, Table 1). Regarding the cellulose synthesis, five genes, including a cellulose synthase and four cellulose-synthase-like genes, were up-regulated in Pixel while two genes coding for two cellulose-synthase-like proteins were down-regulated in Dardo (Figure 3).

In Dardo, observing carbohydrate metabolism, the synthesis of the main tomato fruit sugars appeared limited. In particular, two genes coding for fructose-bisphosphate aldolase (FBA), involved in the biosynthesis of sucrose (from photosynthesis), were down-regulated, including *Solyc02g062340* and *Solyc02g084440* as well as two genes coding for the sucrose-phosphate synthase (SPS) involved in the biosynthesis of sucrose-6-P, the substrate for sucrose synthesis (*Solyc07g007790* and *Solyc11g045110*) (Figure 3). In sucrose degradation, there were two up-regulated genes, a phosphotransferase, belonging to the hexokinase family (*Solyc04g081400*, logFC = 1.12) and a fructokinase (*Solyc03g006860*, logFC = 0.97) (Figure 3) as well as two up-regulated genes coding for the sucrose synthase (Susy, Figure 3). In addition, also the hexokinase *Solyc03g121070* (logFC = 1.79), involved in the degradation of hexoses, was up-regulated. Finally, in Pixel, a conspicuous number of up-regulated genes related to sugar synthesis (glucose and sucrose) confirmed the sweeter flavor of Pixel tomatoes harvested in Battipaglia (Figure 3).

Both genotypes showed DEGs in the amino acid metabolism, (Appendix A). In Dardo, a higher number of differentially expressed genes was found in the aspartate pathway with a preponderance of up-regulated genes. In Pixel, a conspicuous number of genes were involved in the aromatic amino acid synthesis. In Pixel Battipaglia, there was the activation of phenylalanine and tyrosine synthesis with the up-regulation of three genes, *Solyc11g017240*, *Solyc11g066890*, and *Solyc06g050630*, together with a general down-regulation of tryptophan synthesis. Among the most up-regulated genes, we found an arogenate dehydrogenase (*Solyc06g050630*, logFC = 2.24), involved in the biosynthesis of tyrosine and phenylalanine, a proline dehydrogenase (*Solyc02g089620*), implicated in the degradation of proline, and an arginine/ornithine transporter (*Solyc02g077910*, logFC = 2.07). Pixel, also down-regulated in Battipaglia, the isopropylmalate synthase (*Solyc06g053400*) and 3-isopropylmalate dehydrogenase (*Solyc05g009030*) taking part in leucine synthesis. In addition, Pixel Battipaglia down-regulated the threonine deaminase (*Solyc10g083760*) involved in the isoleucine synthesis. Of interest, the two genotypes showed contrasting patterns for arginine biosynthesis but shared the regulation of the glutamate and aspartate production. In Dardo, the down-regulation of the glutamate decarboxylase (*Solyc03g098240*, logFC = −0.467) and of the glutamate synthase (*Solyc03g083440*, logFC = −4.14) and the up-regulation of the glutamine synthetase (*Solyc11g011380*) were observed. In addition, *Solyc07g032740*, belonging to aspartate biosynthesis, was down-regulated. Pixel down-regulated glutamate decarboxylase (*Solyc03g098240*, logFC = −0.73), shared with Dardo, and down-regulated the glutamate synthase (*Solyc03g063560*, logFC = −0.67). In the aspartate metabolism, two genes resulted as down-regulated in Battipaglia (aspartate aminotransferases *Solyc08g041870* and *Solyc07g032740*).

### 2.3. DEGs Involved in Secondary Metabolism

In both genotypes, DEGs between the two environments and implied in the secondary metabolism were mainly related to phenylpropanoids, particularly lignin and flavonoids, and carotenoids metabolism. In both genotypes, the lignin pathway was induced in Battipaglia, although with some differences: while Dardo up-regulated the terminal part of the pathway, Pixel activated the central part. In Dardo, 12 DEGs were mapped, of which 8 genes were up-regulated and 4 genes were down-regulated (Figure 4).

The lignin pathway was activated in both genotype DEGs with higher logFC in comparison to other pathways: for example, in Dardo, among the genes of interest in this pathway, it is important to highlight the CCoAOMT, *Solyc10g050160* (caffeoyl-CoA 3-O-methyltransferase) with a high fold change (logFC = 2.73) (Figure 4). In Pixel, the environment showed a profound effect on the genes involved in the lignin synthesis. Among the 16 genes differentially expressed in Pixel, all resulted as up-regulated (Figure 4). In particular, the gene *Solyc03g097170*, encoding for cinnamoyl-CoA reductase (CCR1), had a logFC = 3.46. Also, for wax metabolism, two genes in Dardo (*Solyc09g065780* and *Solyc11g012260*) and five genes in Pixel (*Solyc02g085870*, *Solyc02g093640*, *Solyc05g009270*, *Solyc07g006300*, *Solyc11g012260*) were up-regulated, respectively. In particular, the epoxide hydrolase, involved in cutin polymerization [11], showed a high fold change (*Solyc05g054330*, logFC = 3.38 in Pixel and logFC = 1.81 in Dardo).

In the frame of phenylpropanoid metabolism, the flavonoid sub-class showed a series of changes in the comparisons under study. Indeed, Pixel showed four up-regulated genes involved in the synthesis of chalcones, including the chalcone synthase (*Solyc09g091510*) with a logFC = 4.94, whereas in Dardo only one gene was involved in this pathway and it was up-regulated and shared with Pixel with a logFC = 2.01.

Another metabolic route challenged by the environment was the carotenoid pathway (Figure 5). More specifically, two genes involved in lycopene synthesis (phytoene desaturase, PDS, and z-carotene desaturase, ZDS) were down-regulated in Pixel, while in Dardo one was up-regulated (z-carotene isomerase, Z-ISO) and one down-regulated (PDS). In addition, in Dardo Battipaglia, we observed the up-regulation of lycopene beta cyclase, *Solyc04g040190* (LCYB), involved in the conversion of lycopene in α- and β-carotene and of the beta-carotene hydroxylase-1, CHY-B1 (*Solyc06g036260*) that allows the conversion of the b-carotene in zeaxanthin. Again, three regulators of the carotenoid synthesis were found to be up-regulated only in Dardo Battipaglia, including STAY-GREEN1 (SGR1, *Solyc08g080090*), the transcription factor NAC2 (*Solyc04g005610*), and ethylene response factor B3 (ERF.B3, *Solyc05g052030*) [12].

### 2.4. Sensory Evaluation

To understand the effect of the environment on the sensorial quality of tomato fruits, a sensory evaluation was performed on each genotype harvested in the two environments. Both tomato cultivars were perceived in a different way when harvested in Battipaglia or Fondi. The results of the triangle test demonstrated for the Dardo pair a significant difference between the two environments (*p* ≤ 0.05). For the Pixel pair, the difference was significant at *p* < 0.01. In total, 90% of the judges who correctly performed the triangle test perceived the Dardo fruits cultivated in Battipaglia as firmer than the fruits harvested in Fondi. The majority of the judges who correctly performed the triangle test (85%) also perceived the Pixel-Battipaglia fruits as firmer than Fondi fruits as well as sweeter and less sour (80% of the judges). On the contrary, the Pixel samples cultivated in Fondi were perceived as having more seeds (50% of judges) compared to the samples cultivated in Battipaglia (Figure 6A). Finally, 95 and 99% of judges, respectively, perceived the Dardo fruits harvested in Fondi as sweeter and with a more intense red color than fruits from Battipaglia (Figure 6B). We also measured the pH and refractometric solid content (°Brix). The pH in both genotypes was not significantly different between the two environments. For Dardo, the pH was 4.31 ± 0.12 in Battipaglia and 4.33 ± 0.07 in Fondi, while for Pixel it was 4.30 ± 0.03 in Battipaglia and 4.31 ± 0.06 in Fondi. The Student’s *t*-test showed significant differences for the °Brix in Pixel, which was 5.83 ± 0.06 in Battipaglia and 7.13 ± 0.06 in Fondi. For Dardo, the °Brix was not significantly different (9.03 ± 0.31 in Battipaglia and 7.03 ± 1.31 in Fondi) between the two environments.

### 2.5. Metabolite Assessment Analysis

In order to assess if transcriptomic changes reflected alterations at the metabolite level, thirty-three key compounds involved in the traits related to sensory perception (sugars, acids, amino acids, flavonoids, carotenoids, etc.) were selected and measured by LC-HRMS in the ripe fruits harvested in the two cultivation areas. First of all, a principal component analysis (PCA) was carried out based on the variation in the levels of the compounds measured in each genotype (Figure 7) to explore the relationship between metabolites and to ascertain their variability in Fondi and Battipaglia. The PCA explained 64.9% of the total variation, showing a clear separation of genotype metabolites between the two environments. The effect of the environment was higher in Dardo than in Pixel, as shown by the wider separation of the samples in Battipaglia and in Fondi. 

Subsequently, the statistical test (*t*-test) showed that 7 metabolites in Pixel had a significant difference between the two environments and 14 metabolites in Dardo (Table 2). In both genotypes, the metabolites belonged to various metabolic classes, playing relevant roles in different quality attributes.

## 3. Discussion

This work was based on the combination of a transcriptomic, sensorial, and metabolomic analysis of the fruit of two tomato genotypes (Dardo and Pixel) cultivated in two different environments. The reprogramming of 4081 DEG genes in Dardo and 3340 in Pixel revealed that the environment has a profound effect on the gene expression of the two tomato cultivars. Despite the two genotypes sharing many DEGs, a high percentage of the genes was specifically expressed in each cultivar. A total of 1245 genes were activated only in the Dardo and 690 in the Pixel, while 51 common DEGs showed the opposite behavior. This result confirmed that the genomic dynamics that shape the response to the environment are genotype specific [10]. In addition, the metabolic analysis showed that the environment affected the levels of metabolites belonging to primary metabolism such as sugar/nucleotide sugars and amino acids as well as to secondary metabolism such as phenylpropanoids and carotenoids. Dardo was strongly affected by the environment, not only at the transcriptome level due to the higher number of DEGs in comparison to Pixel but also at the metabolic level due to the higher number of metabolites that significantly changed between the two localities. Finally, the sensory analysis showed that the environment affects traits such as firmness, higher for both genotypes in Battipaglia, sweetness, modulated oppositely, and color, mainly affected in Dardo.

The tomato fruit firmness is related to its cell wall structure [13,14] and it is among the main quality traits that determine consumer preferences and shelf life [9,15]. In both genotypes, the environment influenced the expression of several genes related to cell wall metabolism as well as the sensory perception of firmness and the abundance of metabolites that are cell wall precursors. The higher firmness perceived in Battipaglia was supported by the up-regulation of genes involved in the pectin, lignin, and hemicellulose synthesis. Notably, the galacturonosyltransferases *Solyc02g089440* (up-regulated in Pixel) and *Solyc07g055930* (up-regulated in Dardo) are both involved in the synthesis of homogalacturonan, an important pectic polysaccharide [16,17] that increases fruit firmness [18]. Many up-regulated genes were also involved in lignin synthesis, a polysaccharide that improves the rigidity of the plant cell wall [18,19]. The hemicellulose metabolism resulted mainly up-regulated in Pixel. It is worth noting a hydrolase (*Solyc01g081060*) was involved in the production of xyloglucan, which was highly activated in Battipaglia. The over-expression of this gene leads to a greater production of xyloglucan and to a minor depolymerization of the same with a consequent increase in the consistency of the cell wall in tomato fruit [20]. The low level of UDP-glucose in Dardo and of UDP-galacturonic acid in Pixel revealed in Battipaglia, could be due to the higher activation of cell wall precursor synthesis in both genotypes in this locality. The up-regulation of the UDP-glucose 6-dehydrogenases (*Solyc02g067080* in Pixel and *Solyc02g088690* in both genotypes) may result in increased cell wall polysaccharide content and firmness [18]. UDP-glucose 6-dehydrogenase is an important enzyme involved in diverting UDP-glucose to cell wall synthesis (mainly hemicellulose) [18,21]. In addition, in Pixel, the low level of UDP-galacturonic acid could be due to the up-regulation of two genes (*Solyc05g050990* and *Solyc12g010540*) that can catalyze the interconversion between UDP-glucuronic acid to UDP-galacturonic acid [22]. The higher firmness perceived by panelists in Battipaglia could be due also to the up-regulation of genes related to cuticle waxes in both genotypes. Waxing retards the rate of moisture loss maintains turgidity and plumpness, and covers injuries on the surface of the commodity [23]. The higher level of *p*-coumaric in Battipaglia in both genotypes and ferulic acid in Dardo could also suggest the incorporation of phenolic compounds in the cutin. These changes may modify both the elastic and viscoelastic behavior of the cuticle, which becomes much stiffer and less deformable [24].

Tomato flavor is mainly composed of taste and volatile aromas, including sugars, acids, amino acids, vitamin C, and various volatiles [25,26,27,28]. In both cultivars, interesting differences in the genes involved in the metabolism of the sugars were found, especially for sucrose and glucose, master regulators of glycolysis/gluconeogenesis, fructose, sucrose, mannose, and starch metabolism in the tomato fruit [29,30]. Dardo showed a more limited content of sugars (glucose/fructose/galactose/mannose/myo-inositol), which resulted in less sweetness in Battipaglia. The down-regulation of the fructose 1,6-bisphosphate aldolase (*Solyc02g062340*) strongly correlated with reduced sweetness [31] and may affect the glycolysis, gluconeogenesis, and Calvin cycle [32]. The activation of the hexokinase *Solyc04g081400* in Battipaglia further supports a process of sucrose degradation and the lower level of glucose detected because the hexokinase is also involved in the conversion of glucose in glucose-6-phosphate [33]. Hexokinase (*Solyc04g081400*) contributes greatly to fruit cell wall biosynthesis in a mutant consistently firmer than the wild type [18]. In addition, the up-regulation of the fructokinase (*Solyc03g006860*), involved in the conversion of fructose in fructose-6-phosphate, could explain the lower level of this sugar in Battipaglia while the up-regulation of myo-inositol oxygenase (*Solyc06g062430*), which converts myo-inositol to UDP-α-D-glucuronate biosynthesis, could explain a lower level of this sugar in Battipaglia for Dardo. In contrast, in Pixel the genes involved in the synthesis of glucose and sucrose resulted as up-regulated in Battipaglia, which in turn could explain the higher sweetness perceived. In addition, the down-regulation of a fructokinase (*Solyc09g011850*) could explain the accumulation of fructose. An important regulator of hexose sugars (*Solyc03g121070*) [31,34,35] resulted as up-regulated in both cultivars but the different contributions of sugar transporters such as *Solyc04g064620* (logFC = 0.60) and *Solyc05g024260* (logFC = 0.68), up-regulated in Pixel, and *Solyc03g007360* (logFC = −3.85), down-regulated in Dardo, could affect the sugar translocation [36,37,38] and the hexose partitioning in the two genotypes that differ in the ratio between the hexoses and the sweet perception.

Significant differences in the fruit content of important amino acids related to flavor attributes such as glutamine, glutamic acid, and aspartate [39,40] were also found between the two environments. The significantly lower glutamate levels found in Battipaglia per Dardo could be due to the down-regulation in this genotype of glutamate synthase (*Solyc03g083440*), which catalyzes the interconversion of glutamine into glutamate, and to the down-regulation of glutamate dehydrogenase (*Solyc01g068210*) interconverting the glutamate in oxoglutarate. Moreover, the lower abundance of aspartate in Battipaglia for both genotypes could be explained by the down-regulation in Battipaglia of aspartate aminotransferases (*Solyc07g032740*) in both genotypes and of aspartate aminotransferases (*Solyc08g041870*) in Pixel.

The tomato fruit color mainly depends on the quantity and quality of the pigments synthesized in the fruit and belongs to two different classes of secondary metabolites, carotenoids, and flavonoids. In Dardo, the different fruit color between Battipaglia and Fondi, apparent from the sensory evaluation, could be explained by the DEGs found in the carotenoid pathway as well as the high level of β-carotene measured in Battipaglia that made the fruit less intense in red color than in Fondi. In Dardo-Battipaglia, in fact, the up-regulation of lycopene beta-cyclase, *Solyc04g040190* (LCY-B), and down-regulation of phytoene desaturase, *Solyc03g123760* (PDS), reduced lycopene production while the up-regulation of b-carotene hydroxylase 1, *Solyc06g036260* (CHY-B1) led to the formation of zeaxanthin. Tomato lines with yellow-orange fruits showed lutein/zeaxanthin accumulation and lycopene reduction in ripe tomato fruits [41]. In addition, the regulation of carotenoid accumulation in Dardo was also suggested by the up-regulation of the transcription factors SlERF.B3 (*Solyc05g052030*) and NAC2 (*Solyc04g005610*) in Battipaglia. *Solyc05g052030* controls fruit ripening through regulating climacteric ethylene production and carotenoid accumulation [12], whereas *Solyc04g005610* is the core regulator of leaf senescence [42] and can directly regulate the gene expression of abscisic acid biosynthesis and affect the pigmentation and softening of tomato fruits [43,44].

## 4. Materials and Methods

### 4.1. Plant Material

Dardo and Pixel, two plum tomato (*Solanum lycopersicum*) varieties with indeterminate habitus and oval-shaped fruits, were grown in two Italian locations, Battipaglia (province of Salerno, SA) and Fondi (province of Latina, LT). Differences between the two locations regarded soil texture, characterized by the predominance of limestone and clay in Fondi and sand in Battipaglia. In addition, there existed differences in chemical parameters such as limestone (6.3 g/kg in Fondi, absent in Battipaglia), exchangeable magnesium (4.8 meq/100 g in Fondi, 2.9 meq/100 g in Battipaglia), exchangeable potassium (3.4 meq/100 g in Fondi, 1.13 meq/100 g in Battipaglia), and exchangeable sodium (0.89 meq/100 g in Fondi, 0.16 meq/100 g in Battipaglia). The cultivars were cultivated during the summer of 2017 following standard tomato procedures used in the area. The locations were characterized by a lower average air temperature (T) and humidity (U) and higher average number of rainy days (R) in Fondi (T = 22.7 °C; U = 51.95%; R = 4.25 days) than in Battipaglia (T = 23.35 °C; U = 57.1%; R = 2.75 days) during the growing season (http://www.ilmeteo.it/portale/archivio-meteo, accessed on 6 August 2017). At the mature stage (the full appearance of red color on the fruit surface), 200 fruits were collected from the intermediate trusses of the plants. The samples obtained for each genotype were used in part to conduct the sensory evaluation while the rest were chopped, divided into replica aliquots, and frozen under liquid nitrogen for storage at −80 °C.

### 4.2. Sequencing and Transcriptomic Analysis

Three biological replicates were analyzed for each genotype in the two environments. Total RNA was extracted from frozen, homogenized, and powdered fruit tomato samples following the protocol previously described [10]. The RNA quality was checked with an Agilent Bioanalyzer 2100 (Agilent Technologies, Santa Clara, CA, USA). Twelve RNA-seq libraries were prepared starting from 2.5 μg of total RNA, obtained from three biological replicates for each cultivar, using the TruSeq RNA Sample Prep Kit v2 (Illumina Inc., San Diego, CA, USA). The cDNA libraries were prepared with TruSeq Stranded mRNA Library Prep kit (Illumina Inc., San Diego, CA, USA) and sequenced by using an HiSeq 1000 (Illumina Inc., San Diego, CA, USA) sequencer according to the manufacturer’s instructions to generate 2 × 75-bp paired-end reads.

The raw data generated by sequencing, after the quality control and trimming, were mapped against the tomato reference genome obtaining raw transcript counts using AIR bioinformatics software (Sequentia Biotech, Barcelona, Spain, https://transcriptomics.sequentiabiotech.com, accessed on 6 September 2017). The raw reads obtained from three biological replicates of each genotype collected in the different experimental conditions were analyzed with Rstudio (R Core Team, 2019). The R package DESeq2 version 1.42.1 [45] was employed to evaluate the reproducibility of the biological replicates through the analysis of principal components (PCA) and the R package edgeR v.3.28.0 to identify gene expression level changes between the conditions under study [46]. A filtering step was conducted to remove poorly expressed genes. Subsequently, the data were normalized to take into account the different sequencing depths of the libraries through the TMM method. Gene expression changes between the two environments for each genotype were calculated and expressed as “log2 fold change” with differences in expression considered significant with a *p*-value “false discovery rate” FDR < 0.05.

### 4.3. Functional Annotation of Differentially Expressed Genes

Differently expressed transcripts were analyzed by the MapMan software version 3.0.6 (http://mapman.gabipd.org/) [47]. The tomato reference genome annotation file was loaded into MapMan together with the lists of DEG for each tomato cultivar via the software Mercator3.6 [48]. To further investigate the role of DEGs, the TomatoCyc of plant metabolic PathwayDatabases was queried. A focus was dedicated to genes involved in fruit quality, such as genes involved in the metabolism of the cell wall, sugars, amino acids, and secondary metabolites.

### 4.4. Metabolic Analysis

Liquid chromatography coupled with high-resolution mass spectrometry with an electrospray ionization source (LC-ESI-HRMS) or with an atmospheric pressure chemical ionization source (LC-APCI-HRMS) was employed for the identification and quantification of the polar (sugars, amino, and organic acids, phenylpropanoids) and non-polar (carotenoids, fatty acids) metabolites contained in each sample using extraction and LC and HRMS parameters as previously described in [49,50]. For the instrumental analyses, 3 biological replicates from each cultivar were used, randomly grouping 10 fruits from at least 75 plants for each experimental condition. To evaluate the difference in the abundance of metabolites, the *t*-test was used between the two environments.

### 4.5. Sensory Analysis

The fruits of each cultivar harvested in the two environments were analyzed by 46 selected judges who carried out the sensory evaluation under white light and in separate booths of the Sensory Laboratory in the Department of Agricultural Sciences (University of Naples, Federico II).

In the first part of the sensory session, each judge performed two consecutive triangle tests [51] to compare each tomato cultivar (Pixel and Dardo) cultivated in the two environments (Battipaglia vs. Fondi). The order of the two tests was randomized between the judges, thus 23 judges evaluated first the Pixel samples, and the remaining 23 judges evaluated first the Dardo samples.

In the second part of the sensory session, the two pairs of samples were again presented but only to the judges that could discriminate between them in the triangle test. They were asked to observe, touch, smell, and taste each sample in the pair and to freely describe the differences between them by writing down each attribute that the sample has as less or more intense than the other sample.

## 5. Conclusions

Transcriptome analysis of tomato fruit revealed that a set of genes were specifically expressed by each cultivar cultivated in different environments. Changes in metabolite levels, affecting the perception of important fruit attributes such as texture, sweetness, and color, were also found. First, the greater consistency perceived in Battipaglia for both cultivars was supported by the up-regulation of genes involved in the synthesis of pectin, lignin, and hemicellulose. Furthermore, the differentially expressed genes challenged different sugar pathways, producing fruits with contrasting sweetness patterns in the two locations. Finally, the regulation of carotenoid accumulation affects pigmentation, especially for Dardo. The results obtained can help understand the combination of genetic and environmental factors that shape the specific characteristics of each cultivar.

## Figures and Tables

**Figure 1 ijms-25-09015-f001:**
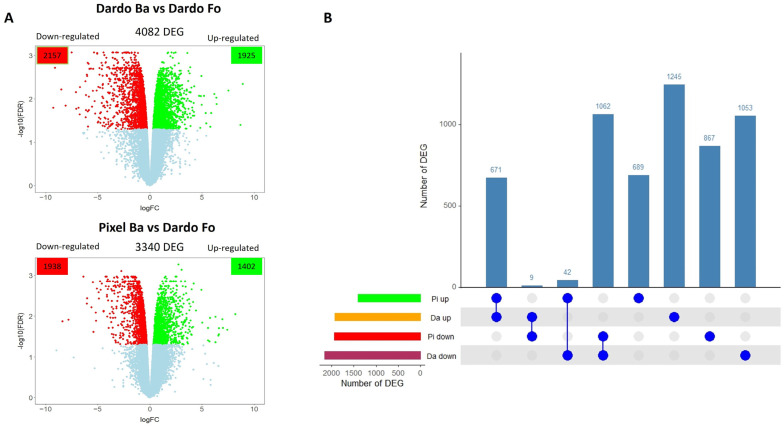
(**A**) Differentially expressed genes (DEGs) identified in two tomato genotypes (Dardo and Pixel) cultivated in two environments. Ba = Battipaglia, Fo = Fondi. (**B**) Intersection of differentially expressed genes. Rows indicate each dataset. Pi = Pixel, Da = Dardo, up = up-regulated, down = down-regulated. Blue filled circle indicates DEG sets participating in the intersection. Vertical bar plots indicate the size of the intersection in terms of the number of DEGs.

**Figure 2 ijms-25-09015-f002:**
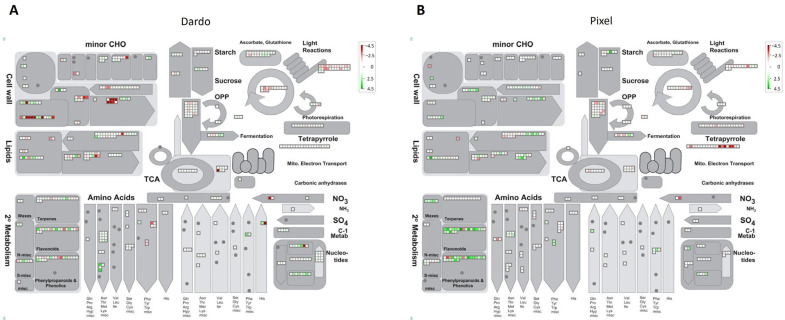
MapMan metabolism overview showing differences in transcript levels between the two environments in Dardo (**A**) and Pixel (**B**). Green squares represent up-regulated transcripts and red squares represent down-regulated transcripts.

**Figure 3 ijms-25-09015-f003:**
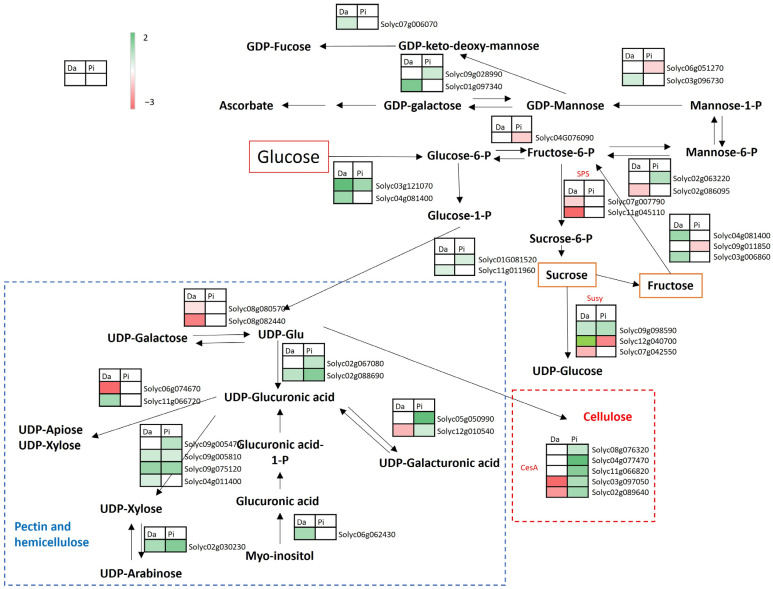
Pixel and Dardo DEGs involved in carbohydrate metabolisms, focusing on cell wall precursors biosynthesis for pectin, hemicellulose, and cellulose. Red boxes indicate down-regulated genes and green boxes indicate up-regulated genes. Da = Dardo, Pi = Pixel.

**Figure 4 ijms-25-09015-f004:**
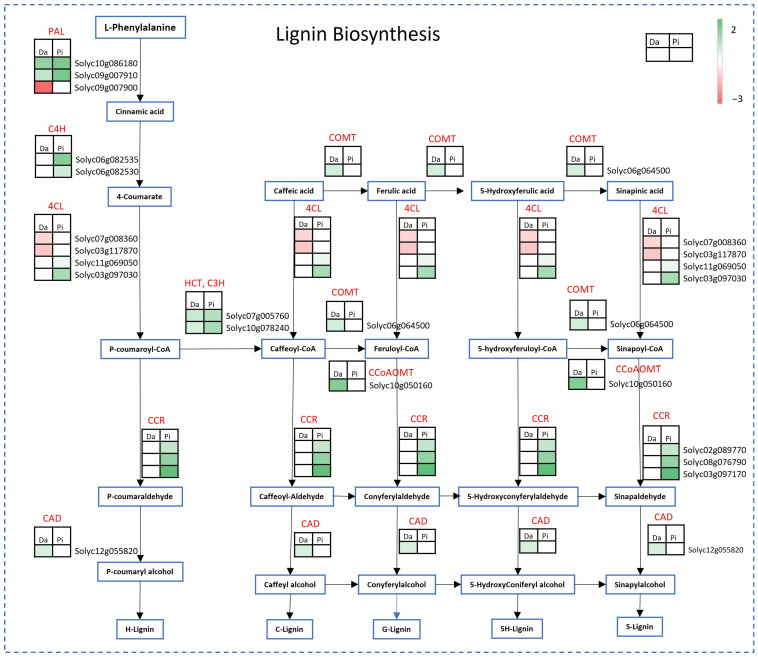
Dardo and Pixel DEGs involved in lignin pathway in tomatoes. Red boxes indicate down-regulated genes, green boxes indicate up-regulated genes. Da = Dardo. Pi = Pixel.

**Figure 5 ijms-25-09015-f005:**
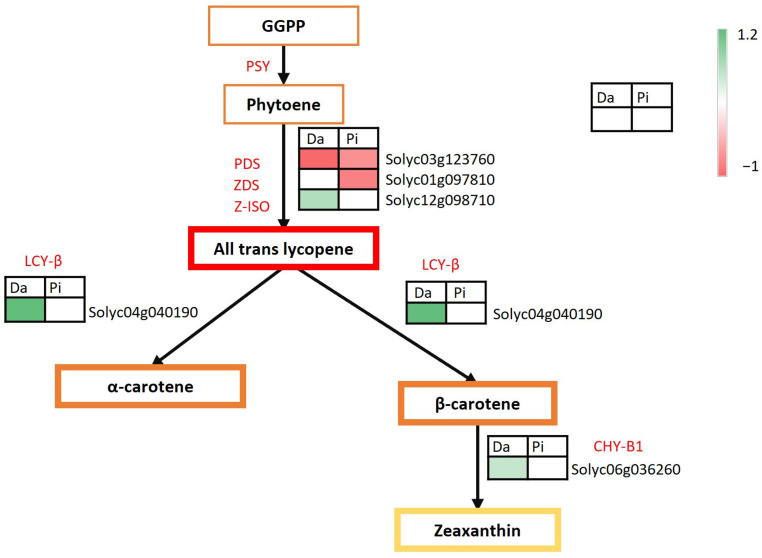
Dardo and Pixel DEGs involved in carotenoid synthesis. Red boxes indicate down-regulated genes, green boxes indicate up-regulated genes. Da = Dardo, Pi = Pixel.

**Figure 6 ijms-25-09015-f006:**
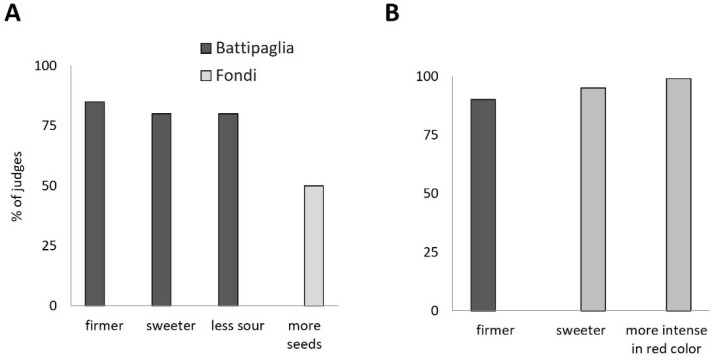
Sensory differences between tomato samples cultivated in Battipaglia and Fondi (**A**) Pixel; (**B**) Dardo.

**Figure 7 ijms-25-09015-f007:**
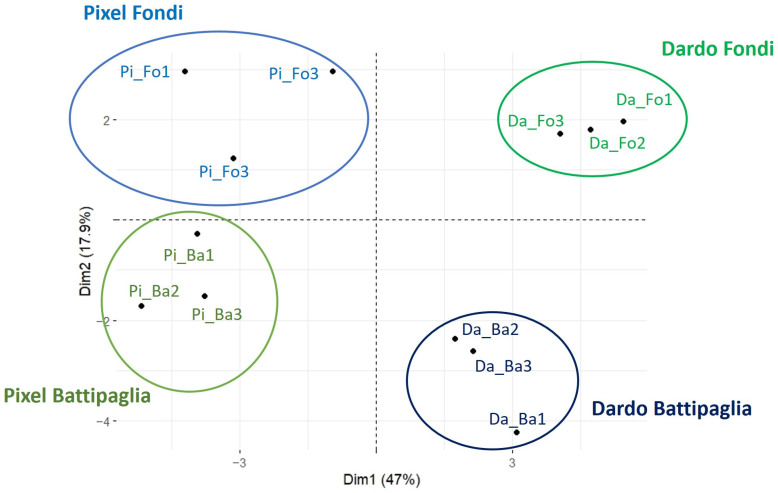
Principal component analysis showing the separation of metabolic profiles for each genotype between the two environments.

**Table 1 ijms-25-09015-t001:** Differentially expressed genes related to fruit quality with contrasting expression in Pixel and Dardo.

Gene ID	Functional Description	Metabolic Pathway	Pixel Ba vs. Fo (logFC)	Dardo Ba vs. Fo (logFC)
*Solyc03g113030*	Aldose 1-epimerase	Glycolysis and gluconeogenesis	−0.50	0.50
*Solyc01g098700*	Branched-chain-amino-acid aminotransferase-like protein	L-leucine biosynthesis/L-leucine degradation I	−0.53	0.53
*Solyc03g097050*	Cellulose-synthase-like protein	Cellulose synthesis	0.82	−3.61
*Solyc03g097050*	Cellulose-synthase-like protein	Cellulose synthesis		
*Solyc02g089640*	Cellulose-synthase-like	Cellulose synthesis	0.92	−2.47
*Solyc01g112000*	Expansin-like protein precursor 1	Cell wall extensibility	1.51	−2.84
*Solyc02g089440*	Glycosyltransferase family protein	Homogalacturonan biosynthesis	0.88	−0.68
*Solyc05g051850*	Putative myo-inositol-1-phosphatase	Myo-inositol biosynthesis	0.91	−0.68
*Solyc01g091840*	UDP-galactose transporter	Nucleotide sugar transport	1.86	−1.22
*Solyc12g010540*	UDP-glucuronate 4-epimerase 4	UDP-α-D-galacturonate biosynthesis I (from UDP-D-glucuronate)	0.63	−1.31

**Table 2 ijms-25-09015-t002:** Mean level of metabolites (±SD, n = 3) collected in Battipaglia and Dardo for the two genotypes. Significant differences between means at *p*-value < 0.05 according to *t*-test are highlighted in bold.

	Pixel Battipaglia	Pixel Fondi	*p*-Value	Dardo Battipaglia	Dardo Fondi	*p*-Value
**Sugars/Sugar alcohol/Nucleotide sugars**						
Glucose/Fructose/Galactose/Mannose/Myo-inositol	**0.4622 ± 0.0310**	**0.6468 ± 0.0937**	**0.0104**	**0.3208 ± 0.0128**	**0.4093 ± 0.0144**	**0.0101**
UDP-Glucose	0.1845 ± 0.0036	0.1963 ± 0.0094	0.3998	**0.1678 ± 0.0049**	**0.2307 ± 0.0087**	**0.0033**
UDP-Galacturonic acid	**0.0038 ± 0.0009**	**0.0078 ± 0.0014**	**0.0116**	0.0043 ± 0.0004	0.0089 ± 0.0022	0.1153
**Amino acids**						
Glutamine	3.4699 ± 0.5620	2.6992 ± 0.5970	0.2490	**6.2400 ± 0.5696**	**4.5354 ± 0.1368**	**0.0437**
Glutamic acid	14.6427 ± 0.3340	14.9509 ± 1.0578	0.4793	**9.5698 ± 0.9092**	**13.0123 ± 0.8218**	**0.0484**
Aspartate	**0.8360 ± 0.0373**	**0.9558 ± 0.0412**	**0.0432**	**0.7022 ± 0.0418**	**0.9397 ± 0.0396**	**0.0146**
1-Aminocyclopropane-1-carboxylic acid	**0.1775 ± 0.0213**	**0.1001 ± 0.0148**	**0.0434**	**0.1591 ± 0.0055**	**0.0708 ± 0.0148**	**0.0051**
**Organic acids**						
Ascorbic acid	0.0629 ± 0.0071	0.0802 ± 0.0117	0.0802	**0.0383 ± 0.0023**	**0.0503 ± 0.0021**	**0.0180**
Shikimic acid	**0.0071 ± 0.0002**	**0.0104 ± 0.0013**	**0.0112**	**0.0055 ± 0.0003**	**0.0078 ± 0.0004**	**0.0112**
**Phenylpropanoids—Phenolic acids**						
*p*-Coumaric acid	**0.0097 ± 0.0005**	**0.0080 ± 0.0010**	**0.0194**	**0.0059 ± 0.0003**	**0.0049 ± 0.0000**	**0.0171**
Caffeic acid	0.0217 ± 0.0004	0.0219 ± 0.0031	0.8907	**0.0119 ± 0.0013**	**0.0168 ± 0.0006**	**0.0284**
Ferulic acid	0.0079 ± 0.0005	0.0123 ± 0.0039	0.0810	**0.0077 ± 0.0010**	**0.0042 ± 0.0002**	**0.0307**
Cinnamic acid	14.2733 ± 0.2453	14.9550 ± 1.0604	0.1039	**9.5698 ± 0.9092**	**13.0123 ± 0.8218**	**0.0484**
**Phenylpropanoids—Flavonoids**						
Dihydrokaemferol	0.0104 ± 0.0029	0.0082 ± 0.0006	0.5234	**0.0196 ± 0.0018**	**0.0076 ± 0.0007**	**0.0032**
**Fatty acids**						
linolenic acid	**0.0230 ± 0.0004**	**0.0413 ± 0.0095**	**0.0217**	0.0355 ± 0.0046	0.0250 ± 0.0028	0.1202
**Carotenoids**						
β-carotene	0.2434 ± 0.0176	0.2689 ± 0.0087	0.5972	**0.4179 ± 0.0205**	**0.3504 ± 0.0023**	**0.0308**

## Data Availability

The sequence data that support the results of this study have been deposited in the public repository Gene Expression Omnibus (GEO) with the following series accession number GSE273560.

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
