# Peer review of "The Impact of Growing Area on the Expression of Fruit Traits Related to Sensory Perception in Two Tomato Cultivars"

_ijms, 2024, doi:10.3390/ijms25169015_

Round 1

Reviewer 1 Report

Comments and Suggestions for Authors

 In the manuscript named “The impact of growing area on the expression of fruit traits related to sensory perception in two tomato cultivars”, Daniela D’Esposito et al have performed RNA-seq and metabolic analysis of tomato between two cultivars, their results have shown different cultivars have different responses to different environments, their findings are helpful for determining tomato fruit development or fruit quality, and which would be useful in future. There were some comments about it.

Major,

(1) Most work is just about RNA-seq analysis, twelve samples, which is not enough for IJMS, please supplement some molecular experiments, such as qRT-PCR for validating RNA-seq results, or physiological biochemical analysis for fruit quality.

(2) Many results were shown with genes expressing profiles description in results section, it was dull, please re-written.

(3) RNA-seq data should be submitted to public database with accession number in manuscript.

Minor,

(4) Figure 1, for volcano map, the limit of y-axis, -log10(FDR), is too small, many genes would be missing with high value (-log10(FDR)). For the upset plot, it would be not suitable in present manuscript, authors can try Venn plot.

(5) The genes in involved in fruit quality, see Table 1, we didn’t know how get it, authors should describe it in detailly in methods section.

(6) The color set would be consisted in manuscript, for example, in DEGs in figure 1, down-regulated genes were set as green, while in MapMan analysis, the green genes were suggested as up-regulated genes, it would cause confusion.

(7) Please remove question mark in figure 5.

Author Response

Thank you for point out several paper issues. Please find below our Point to point response:

(1) Most work is just about RNA-seq analysis, twelve samples, which is not enough for IJMS, please supplement some molecular experiments, such as qRT-PCR for validating RNA-seq results, or physiological biochemical analysis for fruit quality.

The RNAseq technology differently from microarray, is much reliable into the identification of differentially expressed genes because does not suffer of concerns about reproducibility and bias and it is robust enough to validation by qPCR and/or other approaches is not strictly necessary (Coenye, 2021). There are several studies, reported in Everaert et al. 2017, that have specifically addressed the correlation between results obtained with RNA-seq and qPCR. Our experiments were conducted using three biological replicates for sample and the sample error was statistically estimated.

Coenye T. Do results obtained with RNA-sequencing require independent verification? Biofilm. 2021 Jan 13;3:100043.

Everaert C, et al. Benchmarking of RNA-sequencing analysis workflows usingwhole-transcriptome RT-qPCR expression data. Sci Rep 2017;7(1):1559.

We added t physiological biochemical analysis for fruit quality as requested. In particular, we added information about pH value and Refractometric solids content (° Brix). The pH in both genotypes was not significant different between the two environments. For Dardo the pH was 4.31 ±0.12 in Battipaglia and 4.33±0.07 in Fondi, while for Pixel was 4.30 ±0.03 in Battipaglia e 4.31 ±0.06 in Fondi. The student t-test showed significant differences for the Brix in Pixel that was 5.83 ±0.06 in Battipaglia e 7.13 ±0.06 in Fondi. For Dardo the Brix was not significant different (9.03 ±0.31 in Battipaglia and 7.03±1.31 in Fondi).

(2) Many results were shown with genes expressing profiles description in results section, it was dull, please re-written.

We re-written (in some cases eliminated) parts of the results to do much readable

(3) RNA-seq data should be submitted to public database with accession number in manuscript.

We submitted sequence data to the public GEO archive and reported in the manuscript the GEO accession number GSE273560.

Minor,

(4) Figure 1, for volcano map, the limit of y-axis, -log10(FDR), is too small, many genes would be missing with high value (-log10(FDR)). For the upset plot, it would be not suitable in present manuscript, authors can try Venn plot.

The limits of the y-axis of the figure 1 (from 0 to 3) are fitted on the data, even if we increase the limits the visualization of the data do not change in comparison to that we inserted in the manuscript. Here you can see when we increased the scale from 0 to 5. The visualization is similar to that with the limits from 0 to 3.

For what concerns the upset plot, we used it to highlight interesting intersections. Venn and Euler diagram are generally preferred for visualizing diagrams with less than three sets, or when there are only few intersections. Here we have 4 datasets and a total of eight intersections.

(5) The genes in involved in fruit quality, see Table 1, we didn’t know how get it, authors should describe it in detailly in methods section.

We specified in the material and methods the genes investigated and related to fruit quality “A focus was dedicated to genes involved in fruit quality, such as genes involved in the metabolism of cell wall, sugars, amino acids and secondary metabolites.”

(6) The color set would be consisted in manuscript, for example, in DEGs in figure 1, down-regulated genes were set as green, while in MapMan analysis, the green genes were suggested as up-regulated genes, it would cause confusion.

We modified the colors of DEGs in Figure 1 to be consistent with MapMan DEG colors.

(7) Please remove question mark in figure 5.

We uploaded a revised figure 5 without question marks

Reviewer 2 Report

Comments and Suggestions for Authors

Title and Objective: The title is significant, and the study aims to investigate the transcriptional profile of two tomato cultivars (Dardo and Pixel), cultivated in two different areas.

Overall Evaluation: While the title and objective of the study are important, there are several points that need clarification and improvement to make the study valuable for the reader.

Points Needing Clarification and Improvement:

1.     Selection of Cultivars and Areas: Why were the cultivars "Dardo" and "Pixel" chosen? What are the specific conditions of the areas where they were cultivated? It is essential to clarify the reasons why this study is beneficial to the reader, especially regarding the impact of climate changes.

2.     Delay in Publication: The study dates back to 2017, which indicates a significant delay in publication. Given that the research primarily addresses the effects of climate changes, it is crucial to highlight the extent of the changes that have occurred since that period compared to the present.

3.     Harvest Stage: In line 362, you mentioned that the harvest was at the maturity stage. What were the key indicators used to determine this stage for both cultivars?

4.     Citation Error: In line 401, the citation "(Grosso et al., 18; Ahrazem et al., 18)" is unclear and needs correction. Please provide the correct references.

5.     Unclear Sentences: Lines 402 and 403 are unclear and need to be rewritten for better clarity.

6.     Environmental Conditions: The entire manuscript is built on the idea of environmental conditions; however, there is no clear declaration of these conditions. Detailed information about the environmental conditions during the study is necessary.

Comments on the Quality of English Language

Minor editing of English language required

Author Response

Thank for your useful comments to out manuscript. We are providing a response to points that need to be clarified

  1. Selection of Cultivars and Areas: Why were the cultivars "Dardo" and "Pixel" chosen? What are the specific conditions of the areas where they were cultivated? It is essential to clarify the reasons why this study is beneficial to the reader, especially regarding the impact of climate changes.

Dardo and Pixel are very widespread and appreciated cultivars in Italy. Differences between the two locations regarded soil texture, characterized by predominance of limestone and clay in Fondi (Lt) and sand in Battipaglia (Sa). In addition, existed differences in chemical parameters such as limestone (6.3 g/Kg in Fondi, absent in Battipaglia), exchangeable magnesium (4.8 meq/100 g in Fondi, 2.9 meq/100 g in Battipaglia), exchangeable potassium (3.4  meq/100 g in Fondi, 1.13 meq/100 g in Battipaglia) and exchangeable sodium (0.89 meq/100 g in Fondi, 0.16 meq/100 g in Battipaglia).

  1. Delay in Publication: The study dates back to 2017, which indicates a significant delay in publication. Given that the research primarily addresses the effects of climate changes, it is crucial to highlight the extent of the changes that have occurred since that period compared to the present.

We specified in the Materials and Methods, in the paragraph “Plant material”, the climatic conditions of the year of the study.

  1. Harvest Stage: In line 362, you mentioned that the harvest was at the maturity stage. What were the key indicators used to determine this stage for both cultivars?

The key indicators of the maturity stage were the full appearance of red color on the fruits surface. We specified in the manuscript

  1. Citation Error: In line 401, the citation "(Grosso et al., 18; Ahrazem et al., 18)" is unclear and needs correction. Please provide the correct references.

We corrected the error by removing "(Grosso et al., 18; Ahrazem et al., 18)" from the text. The correct references are referred to the numbers [49] and [50].

  1. Unclear Sentences: Lines 402 and 403 are unclear and need to be rewritten for better clarity.

We modify it

  1. Environmental Conditions: The entire manuscript is built on the idea of environmental conditions; however, there is no clear declaration of these conditions. Detailed information about the environmental conditions during the study is necessary.

We specified in the Materials and Methods, in the paragraph “Plant material”, the climatic conditions of the two areas regarding the average air temperature (T) and humidity (U) and average number of rainy days (R).

Reviewer 3 Report

Comments and Suggestions for Authors The cultivation area mainly affects sensory attributes related to texture and flavor and the metabolic pattern of cell wall precursors, sugars, glutamate, aspartate and carotenoids.
That's why it's important to detail:
- Plant material and growing conditions of the two areas
- climatic conditions of the two areas where the two tomato varieties were cultivated
The differences between the two locations:
- soil texture, chemical parameters such as: pH, limestone, exchangeable magnesium
exchangeable potassium, C/N ratio, electrical conductivity, etc
- number of samples obtained for each variety to carry out genes involved in the metabolism of cell wall components, carbohydrates, amino acids and other compounds involved in sensory perception.
In the attachment, you can find the work with all the suggestions that would lead to its improvement.

Author Response

Thank for your useful comments to out manuscript. We are providing a response to points that need to be clarified.

Comments and Suggestions for Authors

The cultivation area mainly affects sensory attributes related to texture and flavor and the metabolic pattern of cell wall precursors, sugars, glutamate, aspartate and carotenoids.
That's why it's important to detail:
- Plant material and growing conditions of the two areas

We specified in the Materials and Methods, in the paragraph “Plant material”, the characteristics of two varieties and  climatic conditions of the two areas regarding the average air temperature (T) and humidity (U) and average number of rainy days (R).

The differences between the two locations:
soil texture, chemical parameters such as: pH, limestone, exchangeable magnesium
exchangeable potassium, C/N ratio, electrical conductivity, etc

We specified in the Materials and Methods, in the paragraph “Plant material”, the difference between the two localities for what concerned the soil properties

number of samples obtained for each variety to carry out genes involved in the metabolism of cell wall components, carbohydrates, amino acids and other compounds involved in sensory perception.

We specified in the Materials and Methods, in the paragraph “Sequencing and transcriptomic analysis” the number of samples.

In the attachment, you can find the work with all the suggestions that would lead to its improvement.

We followed your suggestions for the improvement of manuscript.

Round 2

Reviewer 1 Report

Comments and Suggestions for Authors

Thanks for authors’ work, most of my comments were systematically addressed. But the qRT-PCR is still missing in manuscript, which could be decided by editors or other reviewers, I have no more comments about it. Good luck.

Author Response

But the qRT-PCR is still missing in manuscript, which could be decided by editors or other reviewers, I have no more comments about it. Good luck.Than you for your additional review.

Response: How we explained in the previous revision round there are several RNAseq studies published without qPCR validation. The technology  does not suffer of concerns about reproducibility and bias. Therefore the  validation by qPCR  is not strictly necessary (Coenye, 2021).   We decided to not use it because the results of our biological replicas were statistically robust. I hope that I clarify better our rationale for this choice.

Reviewer 2 Report

Comments and Suggestions for Authors

The authors have made significant improvements to the manuscript. Only minor revisions are needed for further refinement and to align the text with the journal's formatting requirements. Please also reconsider my previous comments to amendments.

Comments on the Quality of English Language

Minor editing of English language required.

Author Response

The authors have made significant improvements to the manuscript. Only minor revisions are needed for further refinement and to align the text with the journal's formatting requirements. Please also reconsider my previous comments to amendments.

Response : We revised the English language, correcting small errors